# A Method for Long-Term Target Anti-Interference Tracking Combining Deep Learning and CKF for LARS Tracking and Capturing

**Tao Zou [1], Weilun Situ [1,2] , Wenlin Yang [2,]*, Weixiang Zeng [1,2] and Yunting Wang [1,2]**

[1] School of Mechanical and Electrical Engineering, Guangzhou University, Guangzhou 510006, China
[2] Guangdong Institute of Intelligent Unmanned System, Guangzhou 511458, China
* Correspondence: yangwenlin@gis.sia.cn

**Abstract:** Autonomous underwater vehicles (AUV) recycling in an underwater environment is particularly challenging due to the continuous exploitation of marine resources. AUV recycling via visual technology is the primary method. However, the current visual technology is limited by harsh sea conditions and has problems, such as poor tracking and detection. To solve these problems, we propose a long-term target anti-interference tracking (LTAT) method, which integrates Siamese networks, You Only Look Once (YOLO) networks and online learning ideas. Meanwhile, we propose using the cubature Kalman filter (CKF) for optimization and prediction of the position. We constructed a launch and recovery system (LARS) tracking and capturing the AUV. The system consists of the following parts: First, images are acquired via binocular cameras. Next, the relative position between the AUV and the end of the LARS was estimated based on the pixel positions of the tracking AUV feature points and binocular camera data. Finally, using a discrete proportion integration differentiation (PID) method, the LARS is controlled to capture the moving AUV via a CKF-optimized position. To verify the feasibility of our proposed system, we used the robot operating system (ROS) platform and Gazebo software to simulate the system for experiments and visualization. The experiment demonstrates that in the tracking process when the AUV makes a sinusoidal motion with an amplitude of 0.2 m in the three-dimensional space and the relative distance between the AUV and LARS is no more than 1 m, the estimated position error of the AUV does not exceed 0.03 m. In the capturing process, the final capturing error is about 28 mm. Our results verify that our proposed system has high robustness and accuracy, providing the foundation for future AUV recycling research.

**Keywords:** autonomous underwater vehicles (AUV); long-term tracking; deep learning; CKF; LARS capturing

## 1. Introduction

AUV applications began in the 1990s. Since then, various types of AUV have been applied to marine science and technology investigations [1–3], seabed exploration [4], underwater pipeline inspection and maintenance [5], resource exploration [6], etc., due to advancements in power enhancement and energy utilization [7]. An AUV is loaded with energy for underwater operations [8], which means that an AUV needs to return to the mother ship in time for energy replenishment, data upload, equipment maintenance, and mission updates [9]. Reliable launch and recycling techniques are essential to ensure the dependability of AUV for recycling [10].

AUV recovery technology can be divided into water recovery and underwater recovery. The recovery of AUVs by watercraft is the current mainstream direction, and a large number of related devices have been successfully used [11,12]. Water recovery can be divided into hanging type and slide type, but the application scope and operability of the hanging type are more robust [13,14]. Therefore, we mainly study the technology of recovering AUVs by

using hanging equipment on the water. In the past, AUV's position in the recovery process of hanging equipment was mainly through human observation, which could have been more reliable and put the staff in a dangerous environment. Therefore, the close-range positioning technology of AUV is essential to research.

Compared with acoustic sensors, GPS, and radar, vision technologies are more suitable for motion measurement during the launch and recovery of AUV with close-range, real-time, and high-precision demand [15–17]. However, there are still relatively few examples of AUV recycling in extreme sea conditions. The position and orientation of AUVs frequently change due to harsh sea conditions. Challenges arise from large appearance variances caused by illumination, deformation, occlusion, and motion [18,19], which can impact tracking, position, and LARS control. The application scenarios of past visual technology to track and locate targets were relatively simple [20]. For example, Wu et al. [21] proposed an autonomous UAV tracking and landing system based on YOLO with high tracking effect and control accuracy levels; however, it depends on the pre-training of the YOLO network [22], making its universality limited. Liu et al. [23] proposed an AUV recovery technology based on the visual framework and verified the feasibility of autonomously recovering AUV using visual technology through experiments. The existing vision technology is mainly for underwater and air environment. There are few visual technologies for a surface ship to recover AUV.

Previous papers and methods have not proposed a system with high robustness, tracking, and real-time performance levels for LARS tracking and capturing [24,25]. To solve the above problems, our proposed system provides improvements for tracking and recycling the AUV in complex marine environments. The key is that the proposed system incorporates deep learning, CKF, and LARS with a complete process and reliable performance. The contributions of our work are:

- We propose the LTAT method to achieve the designated AUV's long-term pixel position information, integrating deep learning and online learning ideas. This effectively solves the inevitable tracking loss problem caused by complex sea conditions during long-term tracking.
- We optimize and anticipate the AUV's position via CKF to lessen the influence of interfering data on position.
- We use pixel position information and binocular camera data to obtain the AUV's coordinates in the camera coordinate system and estimate its orientation. We obtain the relative positions between the end of LARS and the AUV by calculating the coordinate transformation relationship.
- We design the motion trajectory of the end of the LARS using a five-polynomial interpolation method. We use the discrete PID method to control the motion trajectory of the LARS. Based on our proposed system, the complete process of LARS tracking and capturing AUV is virtually verified via a physical simulation system.

The rest of this paper is structured as follows: In Section 2, we describe the framework of the LARS tracking and control system in detail, and we also demonstrate the LTAT method and the CKF in concrete terms. Section 3 shows the simulation of our primary method and system process. In Section 4, we analyze the advantages of our proposed system and its subsequent improvements based on the experimental findings. Finally, we provide our conclusions in Section 5.

## 2. Unmanned LARS Tracking and Capturing System

In this section, we show the complete flow of our proposed system and describe four main modules in detail, including pixel coordinate tracking, relative pose estimation, data optimization, and actuator control.

The system hardware includes the AUV, binocular camera, and three-axis mechanical arm. For the AUV, we selected the Explore-100 produced by Shenyang Institute of Automation Chinese Academy of Sciences for ocean exploration as the experimental model, which has been successfully applied to various sea trial experiments. The system software

framework of this paper is shown in Figure 1. First, the AUV is specified in the image using the binocular camera, and the Siamese region proposal network (SiamRPN) is used to track the specified AUV and output the AUV's pixel position. Tracking has the following two cases: The first is to continuously track the target without interference. The tracking pixel position can be accurately output, and the resulting tracking image output is added to the system database. In the second, when the target has interfered, the YOLO recognizes the potential presence of target regions in the image. The optimal region is filtered using an empirical threshold to update the SiamRPN template frames and pixel positions. Next, the AUV's pose and position are calculated using the binocular stereo-matching principle based on the feature points' pixel positions. Then we used CKF to optimize the positioning data. Finally, using the CKF-optimized position, the LARS is controlled via a discrete PID method to track and capture the AUV.

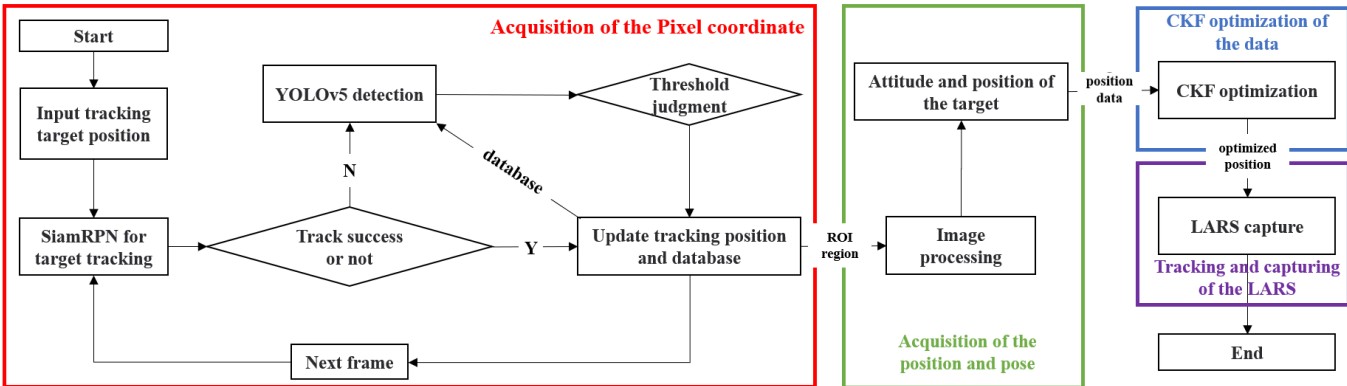

**Figure 1.** LARS tracking and capturing flow chart.

### 2.1. Pixel Coordinates Acquisition of AUV

An AUV's position is the most important data to control LARS. Due to the complex and harsh marine environment, which has a considerable impact on system robustness and position accuracy, we propose the LTAT method for AUV tracking and use CKF to optimize the AUV position.

The SiamRPN provides excellent tracking performance for objects that abruptly change and deform. However, after the target is obscured, it cannot determine the target's position and achieve unmanned re-tracking. The YOLOv5 algorithm has excellent detection speed and accuracy; however, it relies on pre-training and is not effective in detecting targets with deformations and mutations. Combining the advantages and disadvantages of the above two methods, our proposed LTAT method integrates SiamPRN and YOLOv5 with the idea of online learning to achieve highly robust tracking while improving generalizability.

#### 2.1.1. Theoretical Background

SiamRPN is a Siamese network structure based on the RPN [26], which completes target tracking by matching the similarity of the template and test frames [27]. Unlike deep learning-related detection algorithms, it does not require pre-training regarding network structure weights. SiamRPN has universal applicability because it achieves target tracking through end-to-end offline training [28]. Moreover, the algorithm has a simple structure and solves interferences in the tracking process due to illumination, deformation, occlusion, and motion. SiamRPN consists of a Siamese subnetwork and a region-proposal subnetwork, and its network structure is shown in Figure 2. The network outputs the tracked target's offset center coordinate, as well as the tracked area's length and width offset.

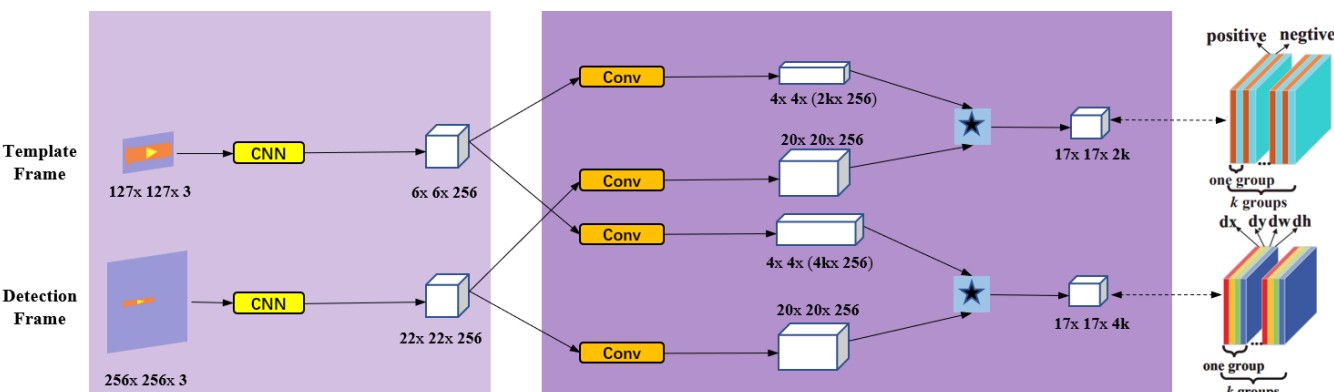

**Figure 2.** SiamRPN main framework: left side is the Siamese network; right side is the region proposal subnetwork; ⋆ denotes correlation operator.

The YOLO series algorithm is a single-stage detection method based on deep learning [29]. It divides the image into cells and predicts the bounding box parameters and confidence, as well as the cell categories [30]. YOLOv5 is the fifth generation update in the YOLO series. It has good detection accuracy and is widely used in engineering applications, where YOLOv5s is the network structure with the fastest operation rate when outputting the optimal predicted target's center coordinates and the boxed region's size.

2.1.2. Long-Term Target Anti-Interference Tracking Method

Because the YOLO relies on network weights trained on large-scale data, SiamRPN does not have a position-update function after target loss, which leads to missing positions when the target is obscured, resulting in a loss of field view. We propose the LTAT algorithm shown in Figure 3 by integrating the SiamRPN's excellent tracking performance with YOLOv5's high-precision detection effects, utilizing online learning ideas to obtain more accurate and robust target-pixel positions.

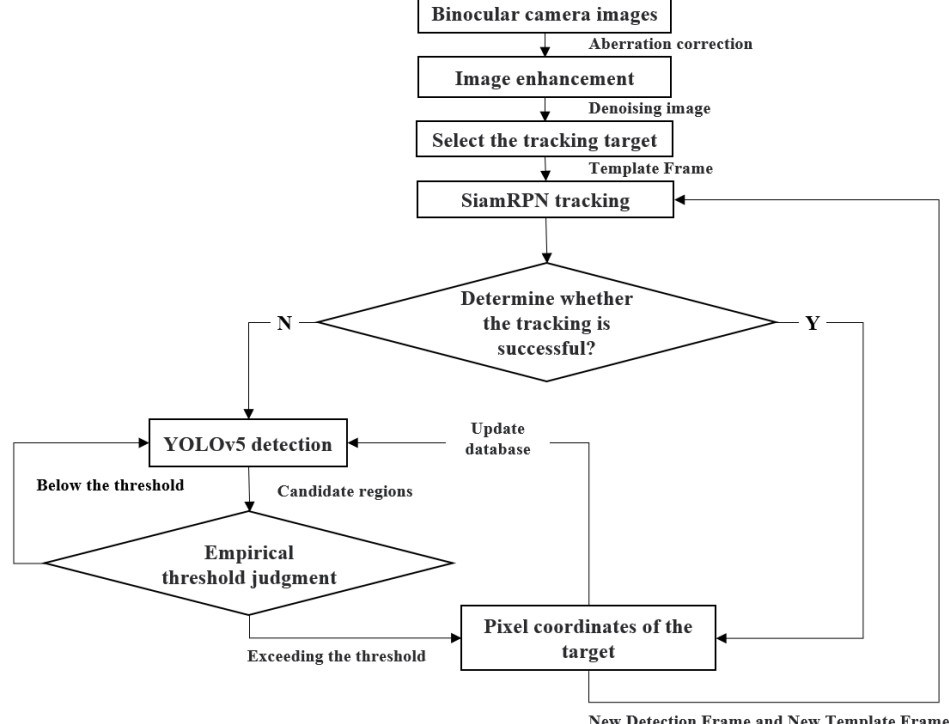

**Figure 3.** LTAT method structure

We used binocular cameras to capture photos; then, we filtered the image noise to improve clarity. Next, we specified the target to be tracked in the image as the template frame. We used the Siamese network to extract features from the template and detection frames, and then we input the features into the RPN's classification and regression branches. The classification branch outputs the category prediction result of each anchor, and the regression branch outputs the center point and bounding box offset of each anchor. Finally, we obtained the center point and bounding box of the tracking target in the detection frame through non-maximum suppression. In order to accomplish the object's continuous tracking effect, we also employed the detection frame's tracking result as a template frame for the following detection frame.

The offline training of the detection algorithm based on the neural network requires a lot of effective data to adjust the network parameters; however, there are few training sets for AUV. In particular, AUVs do not have a unified appearance, and our study observes the AUV from above the water's surface. The presence of the refraction phenomenon led to the AUV's appearance being more unpredictable. All of the above reasons make it impossible to effectively track AUVs through offline training. Therefore, based on the idea of online learning, we utilized the excellent results of the initial tracking experiment for a period of time as system data to train the lightweight YOLOv5 network online. On the one hand, this solved the problem regarding the lack of training data sets. On the other hand, it makes the training data more targeted. Therefore, the online learning method effectively enhanced the system's adaptability and anti-interference performance.

When the tracking target was disturbed and SiamRPN's tracking performance was poor, we used the YOLOv5 to find a potential target region in the image. When the potential target region's score reaches the empirical threshold, output the ideal region and update the SiamRPN tracking model. In conclusion, even when the target is significantly disturbed, the LTAT approach can still track it for a considerable amount of time and determine its pixel position.

### 2.2. AUV Position and Orientation Estimation

The positioning and tracking of AUV is based on three-dimensional space, so it is necessary to obtain the three-dimensional positioning information of AUV. We considered that AUV's recycling needs to achieve close-range high-precision positioning in the marine environment. After analyzing the advantages and disadvantages of various sensors, we found that the binocular camera is the most suitable sensor in the cost, accuracy, and feasibility of the positioning sensors. The binocular camera positioning technology has good positioning accuracy and can realize dynamic positioning in the outdoor environment. Therefore, our research uses the binocular camera to obtain AUV positioning and uses CKF to optimize positioning data in positioning estimation.

### 2.2.1. Coordinate System Conversion Relationship

To control the LARS movement, the relative positions of AUV and LARS need to be calculated in real-time. We set the binocular camera to be installed at the end of the LARS with the lens vertically downward. For the purpose of estimating the relative positions of AUV and LARS, we defined six-coordinate systems, as shown in Figure 4. Set the base of LARS as the world coordinate system $O_w(X_w, Y_w, Z_w)$; camera coordinate system $O_c(X_c, Y_c, Z_c)$; LARS end coordinate system $O_t(X_t, Y_t, Z_t)$; calibration plate coordinate system $O_g(X_g, Y_g, Z_g)$; and the image coordinate system $O_{p_1}(x, y)$ and pixel coordinate system $O_{p_2}(u, v)$. The conversion relationship coordinate system is the key method to estimate relative pose, which is described as follows.

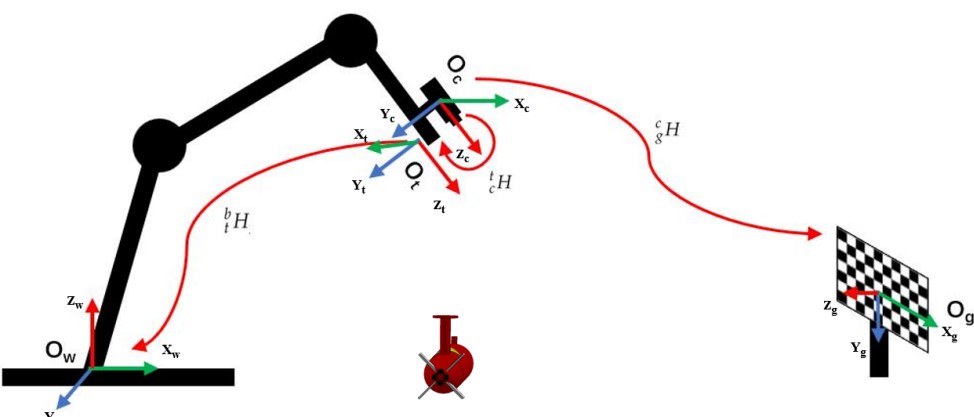

**Figure 4.** Schematic diagram of system coordinate transformation.

We obtained the image pixel coordinates $(u, v)$ of the target using feature point extraction. Generally, the left camera's optical center is set as the origin of the camera coordinate system, and the camera coordinate system is parallel to the image coordinate system $O_{p_1}$. So, we first obtained the conversion relationship from the pixel coordinate system to the image coordinate system. Let dx and dy denote the width and length of the unit pixel, respectively, in the image coordinate system and $(u_0, v_0)$ denote the center point of the pixel coordinate system $O_{p_2}$. The conversion relationship between the pixel coordinate system and the image coordinate system is:

$$\begin{bmatrix} u \\ v \\ 1 \end{bmatrix} = \begin{bmatrix} \frac{1}{dx} & 0 & u_0 \\ 0 & \frac{1}{dy} & v_0 \\ 0 & 0 & 1 \end{bmatrix} \begin{bmatrix} x \\ y \\ 1 \end{bmatrix} \tag{1}$$

We used a binocular camera to obtain the image of the target. The disparity of the feature points in the binocular image can be used to determine the coordinates of the target in the camera coordinate system $O_c$ [31]. We set the coordinates of matching feature points on binocular images to $(x_L, y_L)$, $(x_R, y_R)$. Let f denote the focal length of the left and right cameras and B denote the binocular baseline distance. Furthermore, the conversion relationship between image coordinate $(x, y)$ and camera coordinate $(X_c, Y_c, Z_c)$ is:

$$\begin{cases} X_c = \dfrac{x_L}{x_L - x_R} B \\ Y_c = \dfrac{y_L}{x_L - x_R} B \\ Z_c = \dfrac{f}{x_L - x_R} B \end{cases} \tag{2}$$

The coordinate system's conversion from pixel coordinate system to camera coordinate system is realized above. Because the camera is set on LARS, the conversion relationship between the camera and the LARS end coordinate system needs to be obtained through hand-eye calibration to realize the capturing of LARS. Let $_t^b H$ denote the conversion relationship between the camera and LARS end coordinate system, and let $_g^c H$ denote the conversion relationship between the camera and calibration plate coordinate system, the conversion relationship $_c^t H$ between the camera and LARS end coordinate system is:

$$(_t^b H_j)^{-1} \times {}_t^b H_i \times {}_c^t H = {}_c^t H_j \times (_g^c H_j)^{-1} \times {}_g^c H_i \tag{3}$$

In the above equation, $_t^b H$ can be obtained using external parameters of camera calibration, while $_g^c H$ can be obtained using forward kinematics.

Using the above conversion relationship, the AUV position can be converted from the pixel coordinate system to the LARS base coordinate system. This means that we can locate the AUV according to the feature point information of the AUV and control the LARS tracking and capturing based on the accurate position.

### 2.2.2. Orientation Measurement Method

Feature point extraction and matching of images is a crucial step for the system to position AUV. For the purpose of realizing the function of accurate real-time position, we designed an identifier and adopted the method of sparse stereo matching. Because the triangle's corner-point detection is more distinguished and has fewer feature points, it can meet the demands of practical engineering for tracking and position. Therefore, we designed an isosceles triangle with side lengths of 0.15 m, 0.4 m and 0.4 m on the surface of the AUV as the marker for tracking the target, as shown in Figure 5.

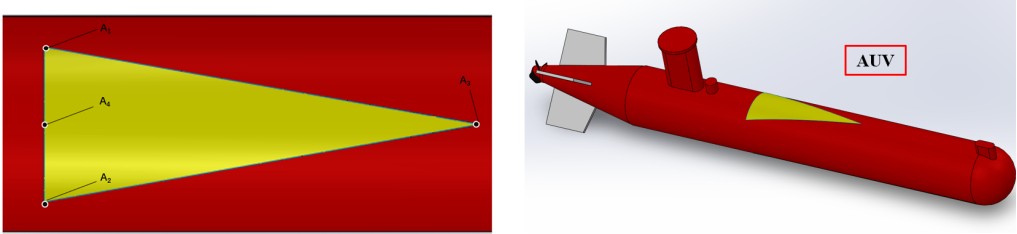

**Figure 5.** Appearance design of AUV.

The appearance of our design is based on the exploration-100 AUV, and its specific physical parameters are shown in Table 1.

**Table 1.** Physical parameters of AUV.

| Weight | Diameter | Length |
|:------:|:--------:|:------:|
| 47 kg | 200 mm | 1.8 m |

The AUV's pitch and yaw angles can be calculated by detecting the target area's three corner points. Let $A_1(x_1,y_1,z_1)$, $A_2(x_2,y_2,z_2)$, and $A_3(x_3,y_3,z_3)$ denote the triangular feature points and $A_4(x_4,y_4,z_4)$ denote the midpoints in the image coordinate system. The triangular feature points can be accessed through the position estimation method, and the midpoint is as follows:

$$\begin{cases} x_4 = \dfrac{x_1 + x_2}{2} \\ y_4 = \dfrac{y_1 + y_2}{2} \\ z_4 = \dfrac{z_1 + z_2}{2} + D \end{cases} \tag{4}$$

where D is the fixed error caused by the marking on the cylindrical surface. The AUV's yaw and pitch angles can be calculated via the midpoint as follows:

$$Pitch = arctan \frac{(z_3 - z_4)}{\sqrt{(x_3 - x_4)^2 + (y_3 - y_4)^2}} \tag{5}$$

$$Yaw = arctan \frac{x_3 - x_4}{y_3 - y_4} \tag{6}$$

In the above equation, Pitch is the pitch angle of the AUV, and Yaw is the yaw angle for straight navigation.

By knowing the pitch and yaw angles, collisions between the LARS and AUV can be effectively avoided during the LARS movement, avoiding unnecessary risk and financial loss.

### 2.3. Position Optimization Based on CKF

There are disturbances, such as wave masking, light changes, and wave fluctuations, in the marine environment. These disturbances can cause errors and offsets in the positioning data. Kalman filter is a widely used data optimization method in the industry [32]. Considering that the AUV's motion system is nonlinear, Kalman filters, such as Extended Kalman (EKF), Unscented Kalman (UKF), and CKF can be used. The specific implementation of EKF is to linearize the nonlinear system [33], which has a large error probability. CKF has a more rigorous and stable theoretical basis than UKF, and when the system state dimension is greater than three, the reliability factor of UKF increases with the increase in dimension, while the reliability and filtering accuracy are greatly reduced. At the same time, the CKF algorithm has higher execution efficiency and lower time complexity, which meets the real-time requirements of our experiment [34–36]. Therefore, position accuracy and anti-interference performance are considerably enhanced in our study by using CKF to optimize and anticipate the position.

We set the position and velocity of the AUV as the state vector of CKF, and the state vector at moment k is formed as follows:

$$\mathbf{X}_k = (x_k, y_k, z_k, v_k^x, v_k^y, v_k^z) \tag{7}$$

where $\eta = (x_k, y_k, z_k)$ is the position of AUV relative to the end of LARS at the moment k, and $\gamma = (v_k^x, v_k^y, v_k^z)$ is the velocity of AUV at the moment k. In our study, we make the assumption that the AUV moves uniformly along the X-axis in a straight line, stays stationary along the Y-axis, and moves in a sine wave along the Z-axis. Consequently, the AUV's state–space model is given as:

$$\begin{bmatrix} x_k \\ y_k \\ z_k \\ v_k^x \\ v_k^y \\ v_k^z \end{bmatrix} = \begin{bmatrix} x_{k-1} + T * v_{k-1}^x \\ y_{k-1} \\ z_{k-1} + C * T * v_{k-1}^z \\ v_{k-1}^x \\ v_{k-1}^y \\ v_{k-1}^z - C * T * z_{k-1} \end{bmatrix} + \mathbf{n}_k \tag{8}$$

where T is the system's sampling interval time, C is determined by the sine wave period, and $\mathbf{n}_k$ is shown as the process noise.

The mother ship obtains the AUV's speed and position information through visual sensors. Let $\mathbf{Z}_k = (x, y, z)$ denote the measurement vector and $\mathbf{m}_k$ denote the measurement noise. The linear observation model is:

$$\mathbf{Z}_k = \mathbf{H}\mathbf{X}_k + \mathbf{m}_k$$
$$\mathbf{H} = \begin{bmatrix} 1 & 0 & 0 & 0 & 0 & 0 \\ 0 & 1 & 0 & 0 & 0 & 0 \\ 0 & 0 & 1 & 0 & 0 & 0 \end{bmatrix} \tag{9}$$

Combing Equations (4)–(6), the discrete dynamic alignment model is written as:

$$\begin{cases} \mathbf{X}_k = \mathbf{f}(\mathbf{X}_{k-1}) + \mathbf{n}_k \\ \mathbf{Z}_k = \mathbf{h}(\mathbf{X}_k) + \mathbf{m}_k \end{cases} \tag{10}$$

where $\mathbf{f}(\bullet)$ is the nonlinear function and $\mathbf{h}(\bullet)$ is the nonlinear measurement transition function. In the non-Gaussian filter estimation process, the key is to solve the multi-dimensional weighted integral [37], which can be formed as follows:

$$I(\mathbf{f}) = \int_{R^n} \mathbf{f}(\mathbf{x})W(\mathbf{x})d\mathbf{x} \tag{11}$$

where $R^n$ is the region of the integration and $W(\mathbf{x})$ is the weight function that $W(\mathbf{x}) \geq 0$. If the solution to the above integral is difficult to obtain, we use numerical integration methods instead. The basic task of numerically computing Equation (10) is to find a set of points $\xi_i$ and weights $\omega_i$ that approximates the integral $I(\mathbf{f})$ by a weighted sum of function evaluations:

$$I(\mathbf{f}) \approx \sum_{i=1}^{m} \omega_i \mathbf{f}(\xi_i) \tag{12}$$

The spherical-radial rule is used by the CKF to determine the points and weights [38]. The dimension of the random variable equals n, and the third-degree spherical-radial rule requires a total of 2n cubature points. The cubature points and associated weights are as follows:

$$\xi_i = \sqrt{\frac{m}{2}}[1]_i$$
$$= \sqrt{\frac{m}{2}} \left\{ \begin{pmatrix} 1 \\ \vdots \\ 0 \end{pmatrix}, \cdots, \begin{pmatrix} 0 \\ \vdots \\ 1 \end{pmatrix}, \begin{pmatrix} -1 \\ \vdots \\ 0 \end{pmatrix}, \cdots, \begin{pmatrix} 0 \\ \vdots \\ -1 \end{pmatrix} \right\} \tag{13}$$

$$\omega_i = \frac{1}{m}, i = 1, 2, \ldots, m = 2n \tag{14}$$

On the basis of the above, the steps of CKF are divided into two parts [39]: prediction and measurement updates. The state variable $\hat{\mathbf{x}}_k$, error covariance matrix $\mathbf{P}_k$, process noise covariance matrix $\mathbf{Q}_k$, and measurement noise covariance matrix $\mathbf{R}_k$ are initialized at first. Then, the state matrix's cubature point is calculated with the equation:

$$\mathbf{P}_k = \mathbf{S}_k \mathbf{S}_k^T \tag{15}$$

$$\mathbf{x}_k^i = \mathbf{S}_k \xi_i + \hat{\mathbf{x}}_k \quad i = 1, 2, \ldots, 2n \tag{16}$$

where $S_k$ is the square root of the covariance matrix. Then, spread cubature points, calculate state prediction, and predict error covariance as follows:

$$\mathbf{x}_{k+1|k}^i = \mathbf{f}(\mathbf{x}_k^i) \tag{17}$$

$$\hat{\mathbf{x}}_{k+1|k} = \omega_i \sum_{i=1}^{m} \mathbf{x}_{k+1|k}^i \tag{18}$$

$$\mathbf{P}_{k+1|k} = \omega_i \sum_{i=1}^{m} \mathbf{x}_{k+1|k}^i (\mathbf{x}_{k+1|k}^i)^T - \hat{\mathbf{x}}_{k+1|k}(\hat{\mathbf{x}}_{k+1|k})^T + \mathbf{Q}_k \tag{19}$$

Here are the procedures for the measurement update section. The equation to recalculate the cubature points and re-spread cubature points will be given as:

$$\mathbf{P}_{k+1|k} = \mathbf{S}_{k+1|k} \mathbf{S}_{k+1|k}^T \tag{20}$$

$$\mathbf{X}_{k+1|k}^i = \mathbf{S}_{k+1|k} \xi_i + \hat{\mathbf{x}}_{k+1|k} \tag{21}$$

$$\mathbf{z}_{k+1}^i = \mathbf{h}(\mathbf{X}_{k+1|k}^i) \tag{22}$$

The measurement prediction, estimate error covariance, and cross-covariance matrix should all be calculated as follows:

$$\hat{\mathbf{z}}_{k+1} = \omega_i \sum_{i=1}^m \mathbf{z}_{k+1}^i \tag{23}$$

$$\mathbf{P}_{k+1}^z = \omega_i \sum_{i=1}^m \mathbf{z}_{k+1}^i (\mathbf{z}_{k+1}^i)^T - \hat{\mathbf{z}}_{k+1}(\hat{\mathbf{z}}_{k+1})^T + \mathbf{R}_k \tag{24}$$

$$\mathbf{P}_{k+1}^{xz} = \omega_i \sum_{i=1}^m \mathbf{X}_{k+1|k}^i (\mathbf{z}_{k+1}^i)^T - \hat{\mathbf{x}}_{k+1|k}(\hat{\mathbf{z}}_{k+1})^T \tag{25}$$

Finally, calculate Kalman gain, update state and error covariance matrix as follows:

$$\mathbf{K}_{k+1} = \mathbf{P}_{k+1}^{xz}(\mathbf{P}_{k+1}^z)^{-1} \tag{26}$$

$$\hat{\mathbf{x}}_{k+1} = \hat{\mathbf{x}}_{k+1|k} + \mathbf{K}_{k+1}(\mathbf{z}_{k+1} - \hat{\mathbf{z}}_{k+1}) \tag{27}$$

$$\mathbf{P}_{k+1} = \mathbf{P}_{k+1|k} - \mathbf{K}_{k+1}\mathbf{P}_{k+1}^z\mathbf{K}_{k+1}^T \tag{28}$$

Loss of vision is a relatively regular occurrence because of the harsh sea environment. Using CKF can prevent interference to position data due to the loss of vision. The anti-interference capabilities of the CKF-optimized position data are substantially enhanced, and it is closer to the true AUV trajectory.

### 2.4. LARS Control

Inverse kinematics can be used to determine the appropriate orientation of the LARS based on the relative position between the AUV and LARS base. We used three-degree-of-freedom LARS in our experiment [40]; therefore, it is necessary to plan the moving trajectory according to the expected pose and calculate the rotation angle, rotation angular velocity, and rotation angular acceleration of each joint. We adopted the PD control method in dynamic control to control the LARS. Through the link parameters, we can calculate the torque M ($q$) required to overcome the LARS acceleration inertia and its influence between different connecting rods, in addition to calculating torque C ($q$), which is required to overcome Coriolis and centrifugal forces, and torque G ($q$) which calculated to overcome the Earth's gravity. The dynamic equation of the LARS is:

$$\tau = M(q)\ddot{q} + C(q, \dot{q}) + G(q) \tag{29}$$

In the above equation, $\tau$ is the control moment vector, q is the rotation angle vector of the system, $\dot{q}$ is shown as the angular velocity vector of the system, and $\ddot{q}$ is shown as the angular acceleration vector of the system.

We used a control method with negative feedback to lessen the inaccuracy levels because LARS's estimated physical parameter values are inaccurate. Our PD control method is given as:

$$\begin{cases} \ddot{e} = K_d\dot{e} + K_p e \\ e = q_d - q \end{cases} \tag{30}$$

In the above equation, e is the position error, $q_d$ is the desired joint rotation angle, and q is the system's control output rotation angle. $K_d$ is shown as the differential coefficient, which can reflect the variation trend of the deviation signal to reduce the control system's oscillation. $K_p$ is shown as the proportional coefficient, which can proportionally reflect the

control system's deviation signal to speed up its dynamic response. Moreover, we assigned no integral coefficient because the LARS is in the dynamic tracking stage [41].

## 3. Simulation and Experimental Results

In this section, we show the results of the experiments we conducted using the LTAT method and position estimation. We verified the robustness of LTAT and the accuracy of position estimation under a variety of sea conditions.

### 3.1. Experimental Design of Simulation

In this paper, we used the ROS framework and Gazebo software to construct the physical simulation environment to realize the modeling, simulation, and visualization of the system's complete process. In the simulation environment, we used LARS as the executive mechanism, and we completed the AUV's model construction and appearance design using Solidworks and software Blender. We used the PyTorch framework and the OpenCV library to implement the LTAT method, where the GPU uses NVIDIA RTX3060. We verified the control method through Simulink on the MatLab platform. Our experimental procedure simulates the mother ship approaching the AUV while maintaining the same forward speed as the AUV; then, it catches the AUV in a reasonably stable scenario. The experimental steps are as follows:

- First, we set LARS to be in an initial state, and we executed the end of LARS to obtain the image data. The image data's simulated scene includes normal driving and bad sea conditions and similar interference. We mainly simulated severe sea conditions for the most common wave coverage interference. By reviewing the relevant literature [42,43], we found that the primary interference in the recovery process was the smaller ships during the simultaneous advance of the two ships, which exists in the direction of heave. Therefore, we set the AUV to perform sine wave movement on the Z axis, make a uniform linear motion on the X axis, and ensure that it is in a fixed position on the Y axis;
- Second, we used a binocular camera to obtain ocean images, and then we used the LTAT method to track the specified AUV in the image. We used the tracking results to calculate the AUV's position, as well as its pitch and yaw angles;
- Finally, we regulated the end of LARS to follow the AUV's position trajectory in accordance with the position results, avoiding collision with the AUV in the process. We operated the LARS to capture the AUV when it sinks on the Z axis.

### 3.2. Long-Term Target Anti-Interference Tracking Experiment

To test the LTAT method, we set the AUV trajectory using Gazebo. We simulated the most common AUV recycling states at sea, including the normal driving and wave-covered states, as well as the interference state of the analogs. To validate the generality of the LTAT, we collected 300 images of AUVs under wave coverage via Gazebo as the tracking data, and we used the labeling software to calibrate the tracking data's true position. Using one-pass evaluation (OPE), we used the first true position as the LTAT method's tracking target. We obtained the LTAT method's success rate and accuracy by comparing the tracking value with the true value. The accuracy and success rates of SiamRPN and LTAT are shown in Figure 6. The LTAT method is superior to the SiamRPN in terms of accuracy and success rate.

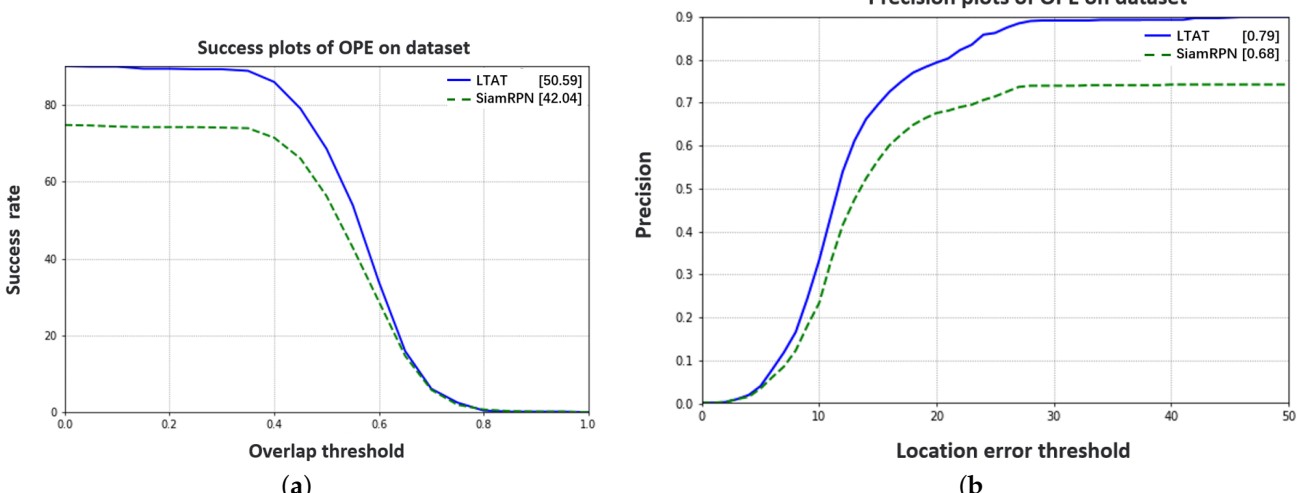

**Figure 6.** Performance comparison of tracking algorithms: (**a**) OPE success rate; (**b**) OPE accuracy.

The visual tracking results for AUVs interfered with by multi-target analogs are shown in Figure 7. The yellow box in Figure 7 is the AUV position obtained through LTAT tracking, and the blue box is the true position of the AUV. Figure 7a and b show the results of the tracking start and end, respectively. The results show that the LTAT method can keep tracking the AUV for a long time when it is disturbed by multi-target analogues.

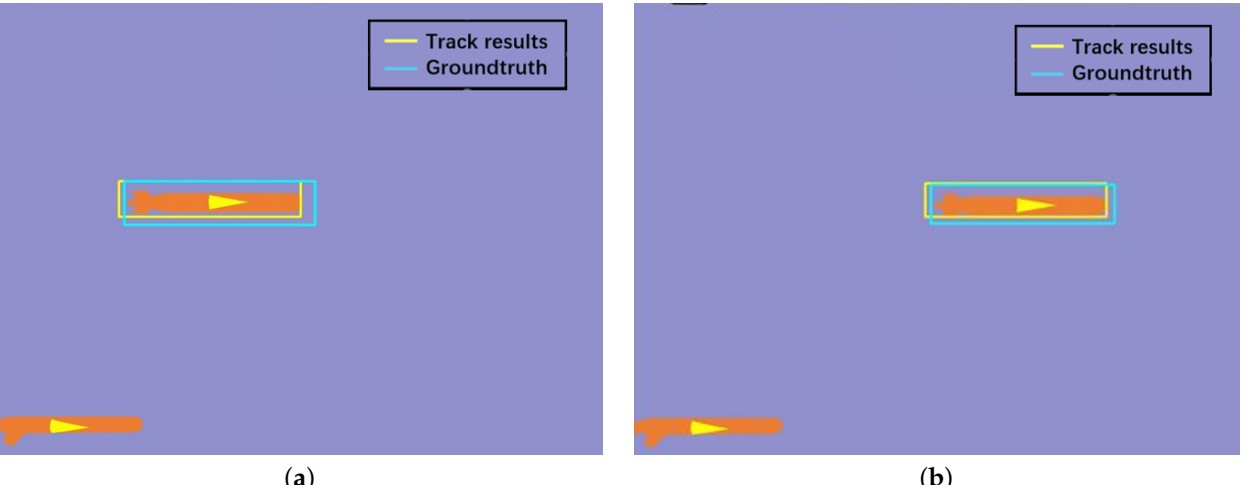

**Figure 7.** Tracking results under interference from similar targets: (**a**) Tracking start frame; (**b**) Tracking end frame.

The visual tracking results of AUVs interfered with by wave cover are shown in Figure 8. Figure 8a shows that the AUV begins to be affected by waves, which will cause the main part of the AUV to sink underwater. Figure 8b shows the subsequent loss of the AUV's visual field under severe wave interference. Figure 8c shows the LTAT method's automatic correction performance for the position after the AUV's main body returns to view. Figure 8 shows that the LTAT method can autonomously correct the position and achieve long-term anti-interference tracking when the AUV is obscured by waves.

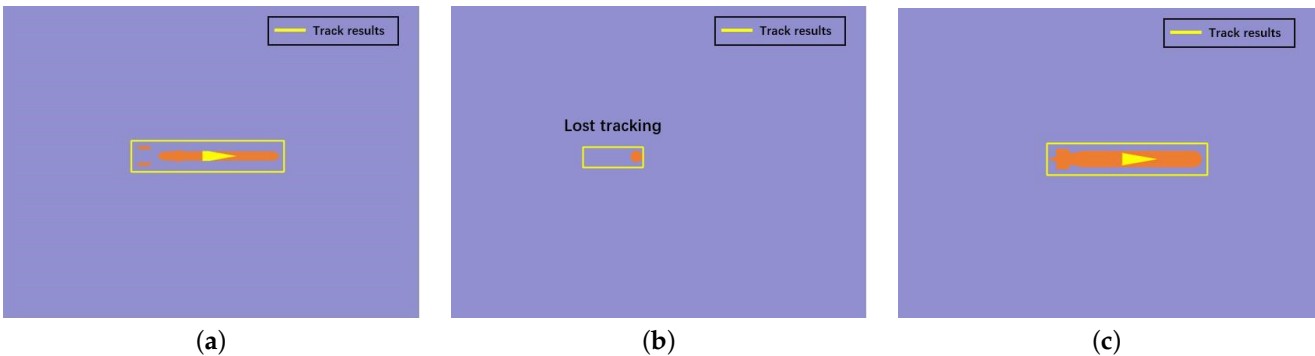

**Figure 8.** Tracking results under interference from waves: (**a**) Start affected frame; (**b**) Tracking lost frame; (**c**) Tracking end frame.

### 3.3. Relative Position and Orientation Estimation Experiments

On the X, Y, and Z axes, we examined trajectory position and position error. We used both severe and normal sea conditions for our experiments. Under normal sea conditions, AUV floats on the water surface without any external factors, and AUV is in an ideal state. AUV is an underactuated model, so it is impossible to independently adjust the depth of the level. To ensure the visual tracking and position of the AUV, we set its buoyancy to be sufficient to support its upper surface above the water surface. The severe sea condition in this paper refers to the situation that AUV is covered by waves. In the experiment, the error we set in the data is caused by the wave cover. Wave masking AUV will lead to an offset of the positioning data.

Figure 9 shows our comparison of the measured, true, and CKF trajectories of AUV motion in a normal sea state. Our comparisons of trajectories for the X and Z axes are shown in Figure 9a,c, respectively. The fact that the three trajectory curves almost overlap demonstrates that there are fewer estimation errors in the X and Z axes. After using CKF, the average value of relative error decreased. Our trajectory comparison of the Y-axis is shown in Figure 9b. There are a lot of similar data in the estimated trajectory and the scatter plot cannot be displayed visually; therefore, we sampled and fitted the estimated trajectory. The position error varies substantially, while the true value of the Y-axis is constant. However, employing the CKF greatly reduced the average position error value.

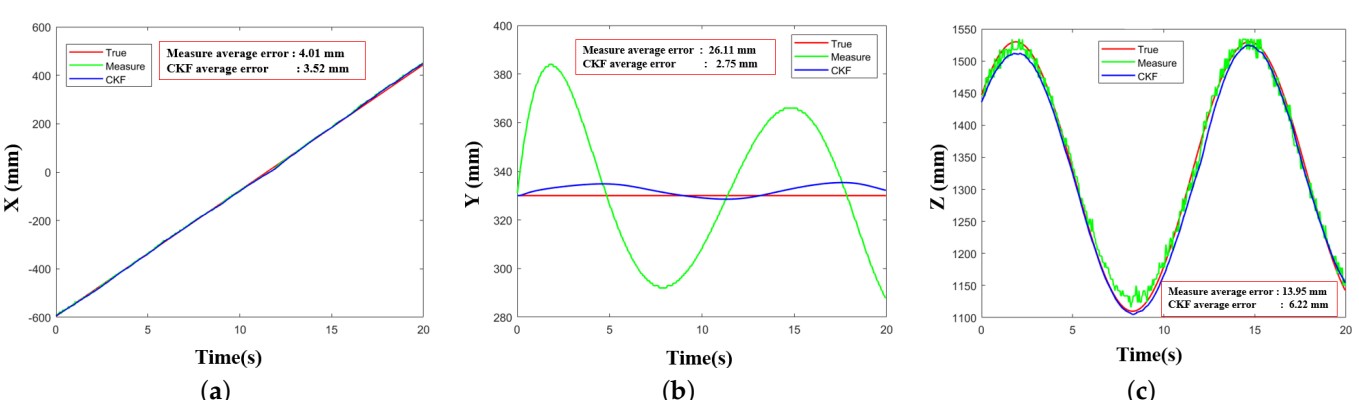

**Figure 9.** Measured, true, and CKF trajectories of AUV motion in normal sea state. (**a**) X-axis, (**b**) Y-axis, and (**c**) Z-axis trajectory comparisons.

To make the position error more intuitive, we calculate the average position error of every 25 data and show the size and trend of the error through a histogram. The error histogram of the X-axis is shown in Figure 10a, the maximum measurement error is 30 mm, and the maximum error after using CKF is 20 mm. Figure 10b is the error histogram of the Y-axis, the maximum measurement error is 50 mm, and the maximum error after using CKF is 5 mm, which has been greatly improved. Figure 10c is the Z-axis error histogram,

the maximum measurement error is 20 mm. After utilizing CKF, the maximum error remains unchanged, but the overall error value is decreased.

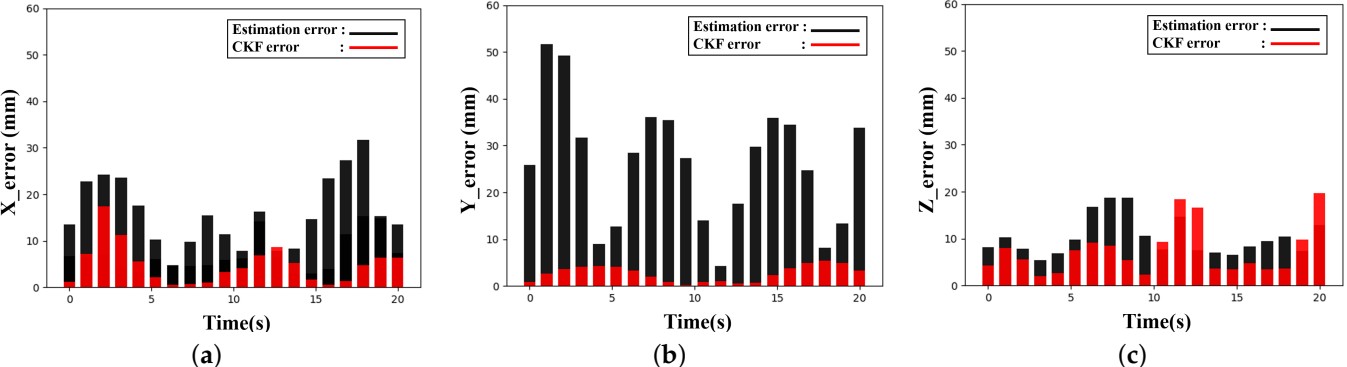

**Figure 10.** Error between true and estimated position in normal sea state: (**a**) X-axis, (**b**) Y-axis, and (**c**) Z-axis error comparisons.

Figure 11 shows the comparison of measured trajectory, true trajectory, and CKF trajectory of AUV motion in severe sea conditions, which demonstrates that the position is seriously disturbed due to the influence of sea conditions. However, when the position is disturbed, the use of CKF can effectively suppress the influence of interference data. As shown in Figure 11a–c, CKF can efficiently filter the interference point and properly estimate the position of the AUV when the position is disrupted.

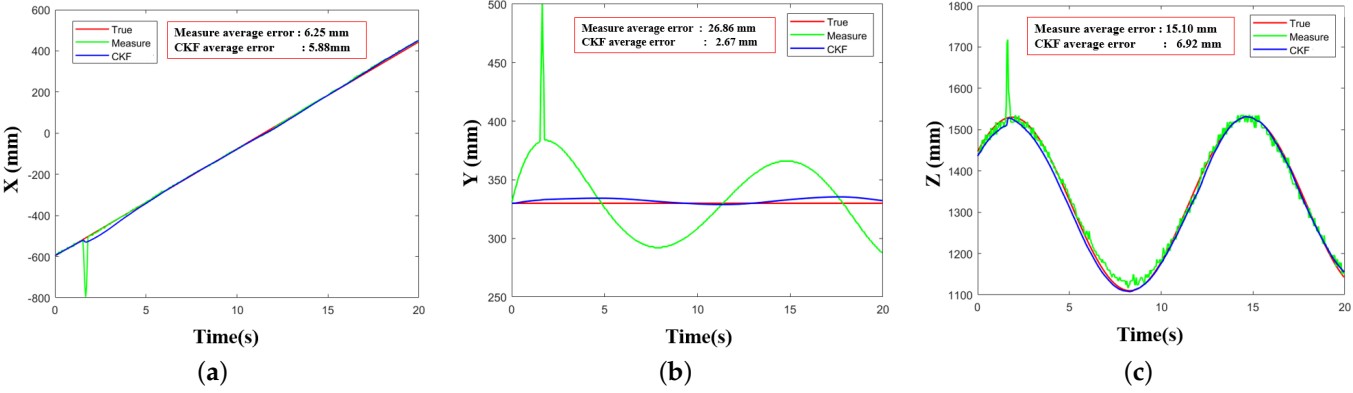

**Figure 11.** Measured, true, and CKF trajectories of AUV motion in severe sea state. (**a**) X-axis, (**b**) Y-axis, and (**a**) Z-axis trajectory comparisons.

The error histogram of the measured and CKF positions in severe sea conditions is shown in Figure 12. Tough sea conditions have a great influence on the measurement position. After using CKF, the interference position error considerably improved on the X, Y, and Z axes. The CKF's optimization effect is consistent with the prior in a normal sea state. Figures 11 and 12 can be compared to demonstrate how effectively CKF inhibits interference positions with large mistakes.

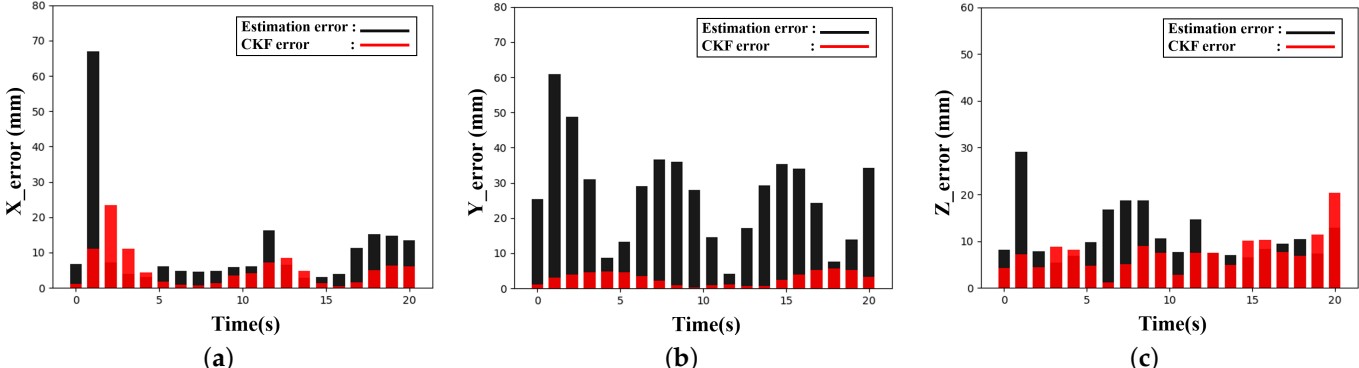

**Figure 12.** Error between true and estimated position in severe sea state: (**a**) X-axis, (**b**) Y-axis, and (**c**) Z-axis error comparisons.

Figure 13 shows a more intuitive three-dimensional position trajectory. Figure 13 shows that there is a non-negligible deviation between the measured and true trajectory. Figure 13a is the trajectory comparison under normal sea conditions, and the error of the measured trajectory is improved after using CKF. Figure 13b shows a trajectory comparison under severe sea conditions, in which the interference points of the measured trajectory are effectively suppressed after using the CKF. The CKF trajectory basically coincides with the true trajectory, showing that our position estimation has high accuracy and strong robustness.

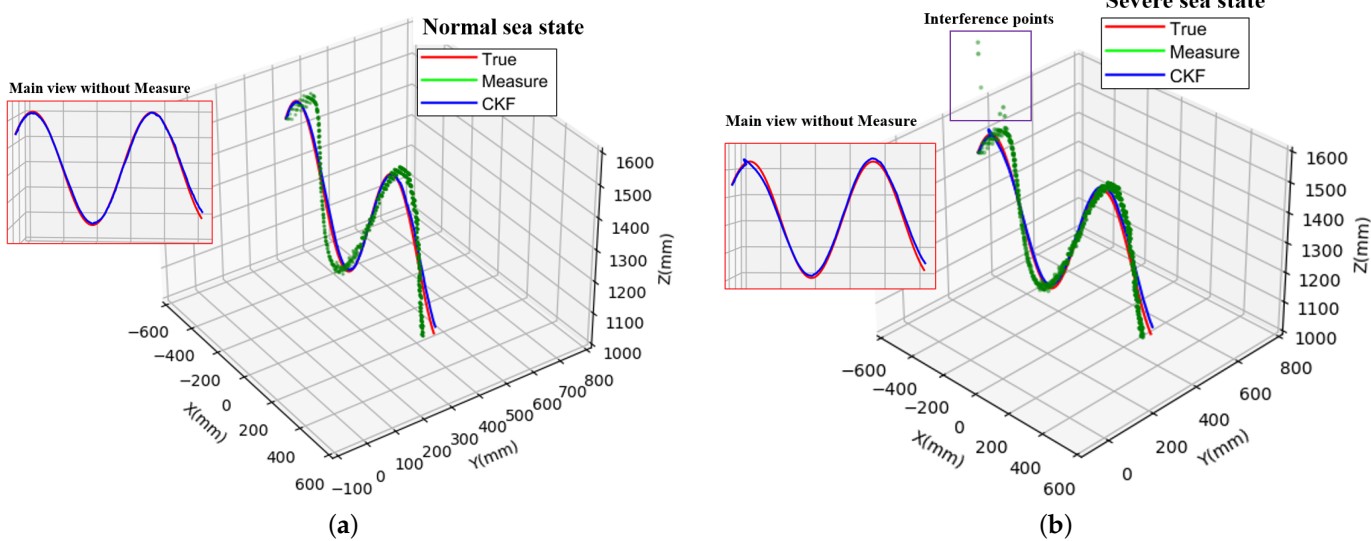

**Figure 13.** 3D trajectories of AUV motion in different sea state: (**a**) 3D trajectory under normal sea state; (**b**) 3D trajectory under severe sea state.

We collected images of the AUV at multiple pitch and yaw angles. The AUV's partial orientation measurement results are shown in Tables 2 and 3, showing that our method can accurately estimate the AUV's pitch and yaw angles.

**Table 2.** Estimates and true values of AUV pitch angles.

| Set the Angle | 15 | 20 | 30 |
|---|---|---|---|
| Pitch angle | 14.7891 | 20.8188 | 33.4522 |

**Table 3.** Estimates and true values of AUV yaw angles.

| Set the Angle | −30 | 0 | 30 |
| --- | --- | --- | --- |
| Yaw angle | −29.5647 | −0.0124 | −29.6427 |

### 3.4. LARS Tracking Trajectory

Because the position trajectory under severe and normal sea conditions is basically the same after using the CKF, we used the positioning trajectory under normal sea conditions as the LARS tracking target. We set the LARS motion to an initial state to be close to the AUV before the LARS starts tracking. Considering that the LARS motion rate and reaction time are lower than the frame rate of visual tracking, we sampled the trajectory for a short time. We performed fifth-order polynomial interpolation trajectory planning between the sampling points to obtain the desired LARS motion trajectory. We obtained the expected rotation angle of each joint by inverse kinematics. Then, we input the expected rotation angle into the control system as the control target; therefore, we obtained the true LARS motion trajectory. In the true control, we set the control system parameters $K_p = 160$ and $K_d = 20$ according to experience, under this parameter control effect is best.

The comparison of the positioning trajectories on the three axes during LARS tracking is shown in Figure 14. During the tracking process, we kept the LARS end approximately 0.8 m above the AUV to avoid collisions. Furthermore, we controlled the LARS to approach the AUV on the X and Y axes to ensure that the AUV is within the capturing range of the end actuator. Then, we analyzed the AUV's acceleration on the Z axis. When the AUV began to sink according to the positioning trajectory, we controlled the LARS to quickly approach the AUV on the Z-axis and complete the capture.

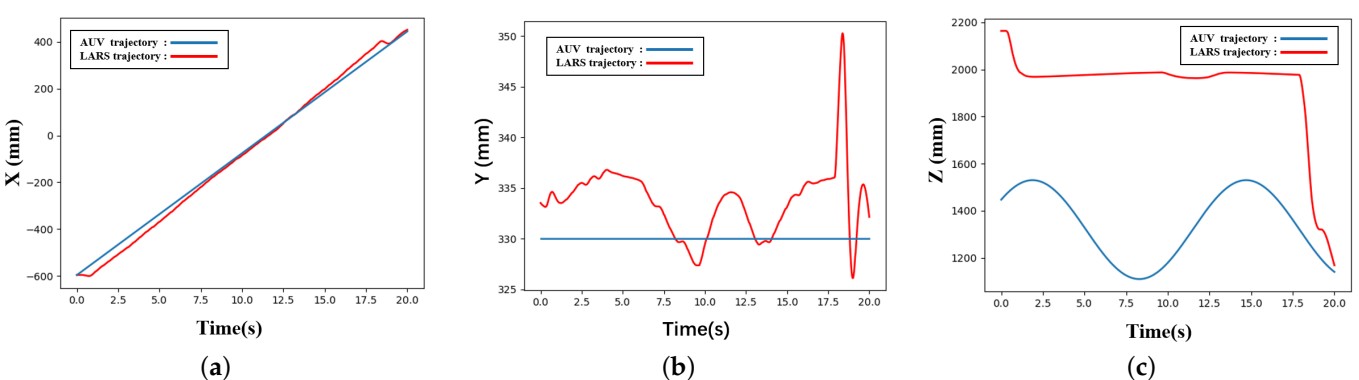

**Figure 14.** Trajectory comparison of AUV and LARS end position in normal sea state: (**a**) X-axis, (**b**) Y-axis, and (**c**) Z-axis trajectory comparisons.

The relative distance between the LARS end and the AUV on the X, Y, and Z axes is shown in Figure 15. Figure 15a shows that the maximum control error on the X-axis is 45 mm. However, because the AUV advances at a constant speed on the X-axis, and because of the AUV's cylindrical shape, the error on the X-axis has little effect on the capturing. The positioning errors on the X and Y axes also decreased during the capture process, and the maximum control error on the Y axis is less than 22 mm as shown in Figure 15b. In the tracking stage, because LARS is kept far from the AUV on the Z-axis, we primarily took its final capturing error into account. Figure 15c shows that the final control error on the Z-axis is about 28 mm. The inaccuracies in the three tracking-stage dimensions and the capture stage are within the real engineering requirements.

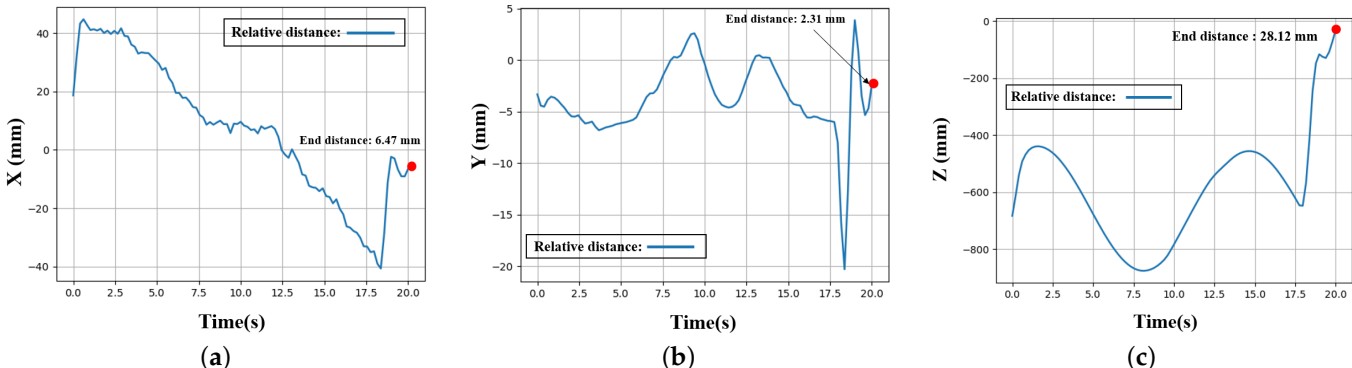

**Figure 15.** Relative position between AUV and LARS end in normal sea state: (**a**) X-axis, (**b**) Y-axis, and (**c**) Z-axis relative position.

The results of the three-dimensional visualization process are shown in Figure 16. When the AUV drops on the Z-axis, the LARS is controlled to quickly approach the AUV and capture it through the recycling device on the LARS end. Because the AUV shows a descending trend, the LARS will not collide with the AUV during the catching process. The control inaccuracy is roughly 30 mm when the capture is finished. Our experiments show that the system in this paper can accurately recycle the AUV in various sea conditions.

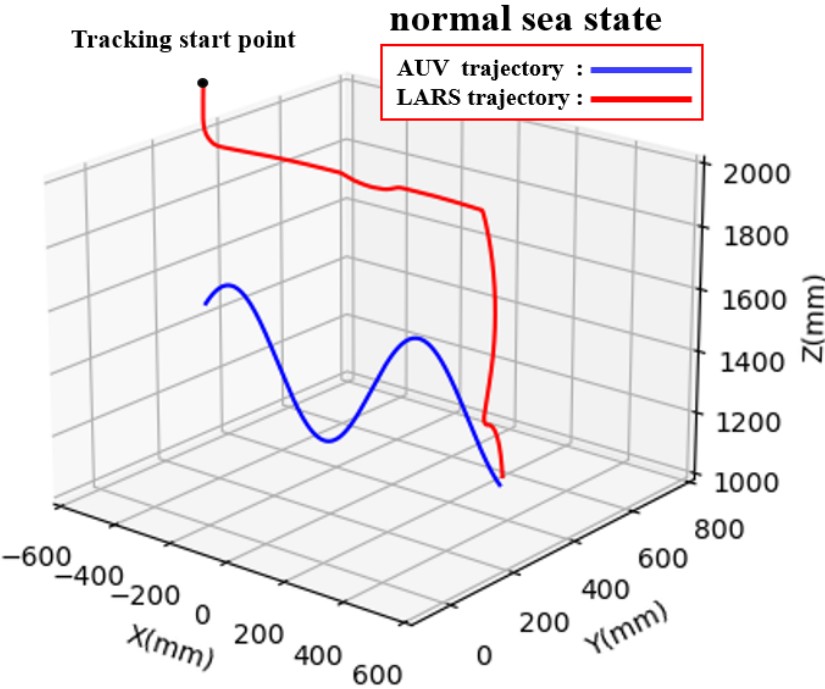

**Figure 16.** Three−dimensional tracking trajectory of LARS.

## 4. Discussion

The simulation results show that the system can recycle AUVs in extreme sea conditions. Compared with the previous AUV recycling system, our proposed system is based on the LTAT method, CKF, and LARS has a better effect on position accuracy and anti-interference. In this section, we analyze the system's tracking effect, position accuracy, and control method based on the experimental data. Moreover, we provide recommendations for improvement after performing an analysis of the influencing factors on the system's performance.

### 4.1. Performance Analysis of LTAT Method

When there is similar target interference and visual loss, the LTAT method can still accurately track the AUV. The LTAT method can swiftly and autonomously identify the AUV's pixel position without manually correcting the tracking error, especially when it reappears elsewhere in the image after missing its field of view due to wave interference; therefore we verified the feasibility of updating the database to track non-standard objects step by step. Furthermore, the LTAT method's tracking rate can reach 48 frames per second at an image resolution of $1280 \times 960$, which is much higher than the 25 frames per second required for real-time tracking.

The deployment platform determines how frequently an online learning module is updated. On the platform in our experiment, the YOLO network's weight updates takes roughly 20 s. The learning module is mainly designed for long-term tracking, and its effect cannot be fully reflected in instantaneous tracking.

The visual sensor's imitations make it impossible to improve the interference position caused by the loss of the visual field from visual technology. Multi-sensor fusion can be utilized for position to make up for the limitations of visual sensors in order to address the position impact brought on by interference.

### 4.2. Analysis of Position Estimation Method

We used binocular images to measure the AUV's position; then, we used a CKF to optimize the measurement position. We carried out an orientation estimation experiment of AUVs in multiple orientations. The position errors on the X and Z axes were smaller when assessing AUV position through binocular pictures; however, the position error on the Y-axis is bigger, as indicated in Figure 9. On the one hand, because the AUV is in a fixed position on the Y-axis when the pixel position changes slightly, it will have a greater impact on the position error of the Y-axis. On the other hand, the position error on the Y-axis is due to floating errors in the feature point coordinates. According to Equation (2), the camera coordinate system's Y-axis position is related to the pixel coordinate system's Y-axis position; therefore, the error on the Y-axis is significant.

It is critical to optimize the AUV's position because its measurement position is strongly affected by interference. In our experiment, we used a CKF to optimize the position, providing a position with stronger anti-interference and higher precision. The CKF can effectively suppress the interference of position with large mistakes, as demonstrated in Figure 11. We optimized the average position error to be below 10 mm, and the maximum position error is optimized to be below 20 mm after using the CKF.

In summary, the factors affecting position accuracy include the extraction method of feature points, external interference factors, and image resolution. A more reliable feature point extraction technique can result in a more precise position. In order to increase position accuracy, the Kalman filter can also be used to combine data from the multi-sensor.

### 4.3. Constraint and Improvement of Control Method

Between the LARS end trajectory and the true AUV trajectory, there is a divergence. One reason is that there are unavoidable measurement inaccuracies in the measured position. On the other hand, as our work makes use of a true physical model, there will be a variety of disturbances in the LARS control. The control method in this paper is to reduce the influence of these disturbances on the end of the LARS. The control error of LARS is about 2% relative to the measurement position and about 5% relative to the true position. This error is acceptable in the capturing process because the capturing range of the capturing mechanism at the end of LARS is wide enough for AUV recovery. In our experiment, it is possible to reliably capture an AUV through LARS.

However, our system does not consider the influence of waves on LARS, although it can compensate for the influence of waves by a six-degree-of-freedom parallel manipulator platform to ensure that LARS is in a relatively stable state. LARS's motion can also be managed using a more sophisticated control method to compensate for wave interference.

## 5. Conclusions

Optimizing the positioning and capturing of AUV recycling system are the main objectives of our research. We construct a complete AUV recycling system in our experiment. The system can adapt to the challenging marine environment and complete the positioning and capturing of the specified AUV via LARS under various disturbances.

Specifically, we used the LTAT method, CKF, and LARS to achieve the specified AUV positioning and capturing. At the same time, we calculated the AUV's yaw and pitch angles by using position data. Our experiments demonstrate that our proposed system can achieve an AUV position error of less than 0.02 m and a control error of LARS of less than 0.03 m when the AUV fluctuates with waves at 0.2 m; furthermore, the initial relative distance between AUV and LARS does not exceed 1 m. This satisfies the performance criteria for precise and real-time AUV recovery.

However, our proposed system still has room for improvement. In our research, our proposed system is still in the laboratory research stage and cannot be verified by sea trials. Subsequently, we plan to verify the system in the sea and develop improvements during the experiment. By continuously improving the sensor fusion and LARS control methods, as well as the mechanical structure of the actuator, an autonomous and intelligent AUV unmanned recycling system can be realized in the future.

**Author Contributions:** Conceptualization, T.Z., W.S. and W.Y.; methodology, T.Z., W.S.; software, W.S. and W.Z.; validation, W.S.; formal analysis, T.Z., W.S. and W.Y.; investigation, T.Z., W.S., W.Y., W.Z. and Y.W.; resources, T.Z.; data curation, W.S.; writing—original draft preparation, T.Z., and S.S.; writing—review and editing, T.Z., W.S., W.Y., W.Z. and Y.W.; visualization, W.S.; supervision, W.Y.; project administration, T.Z. and W.Y.; funding acquisition, T.Z. and W.Y. All authors have read and agreed to the published version of the manuscript.

**Funding:** This work was funded by the National Natural Science Foundation of China [Grant number 52171331]; the International Science and Technology Cooperation Project of Guangdong Province [Grant number 2022A0505050027]; Guangdong Basic and Applied Basic Research Foundation [Grant number 2021A1515012552].

**Data Availability Statement:** The data that support the findings of this study are available from the corresponding author upon reasonable request.

**Conflicts of Interest:** The authors declare no conflict of interest.

## Abbreviations

The following abbreviations are used in this manuscript:

| | |
|---|---|
| AUV | Autonomous underwater vehicles |
| CKF | Cubature Kalman filter |
| EKF | Extended Kalman filter |
| GPU | Graphics processing unit |
| INS | Inertial navigation system |
| IMU | Inertial Measurement Unit |
| LARS | Launch and recovery system |
| LTAT | Long-term target anti-interference tracking |
| OPE | One-pass evaluation |
| PID | Proportion integration differentiation |
| ROS | Robot operating system |
| SiamRPN | Siamese region proposal network |
| UKF | Unscented Kalman filter |
| UWB | Ultra-Wide Band |
| VIO | Visual-Inertial Odometry |
| YOLO | You Only Look Once |

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
