# Peer review of "A Method for Long-Term Target Anti-Interference Tracking Combining Deep Learning and CKF for LARS Tracking and Capturing"

_remotesensing, doi:10.3390/rs15030748_

Round 1
Reviewer 1 Report
This paper only carried out the simulation verification, lacking the experimental link. The experimental verification is very important for the verification of the paper results. It is suggested to add this part.
Author Response
Dear Reviewer,
We feel great thanks for your professional review work on our article. As you are concerned, there are several problems that need to be addressed. Responses to specific questions are uploaded in the documentation below. Best wishes to you.
Sincerely,
The Authors

Reviewer 2 Report
The authors have proposed the a long-term target anti-interference tracking method that integrates deep learning and online learning for unmanned recycling of AUVs in underwater environment. I paper is written well and the results are interesting. I do have some minor comments that should be addressed before the paper can be accepted.
1. How are the hydrodynamic coefficients of AUV determined for modeling the dynamics of AUV. There are both simplified analytical methods and and high-fidelity computational fluid dynamics simulations that has been used in the literature to compute the hydrodynamics coefficients. Which method is used in the present study?
2. The authors have demonstrated the results in a simulation environment. Do authors have any plan to validate the proposed method in actual field trials? What challenges do authors anticipate while applying the proposed method in real-world settings?
3. Is there any reason behind using cubature Kalman filter instead of the plain vanilla version of the Kalman filter?
4. The online learning is not clarified in the text. What is the difference between online learning and offline training of neural networks?
Author Response
Dear Reviewer,
We feel great thanks for your professional review work on our article. As you are concerned, there are several problems that need to be addressed. According to your nice suggestions, we have made extensive corrections to our previous draft, the detailed corrections are listed below.
Sincerely,
The Authors

Reviewer 3 Report
To solve the problem of autonomous underwater vehicles (AUVs) unmanned recycling in complex marine environments, the authors propose along-term target anti-interference tracking method combining deep learning and CKF for LARS tracking andcapturing.To verify the feasibility of the system in this paper, the authors used the robot operating system (ROS) platform and Gazebo software to simulate the system for experiments and visualization.However, there are still some problems in the manuscript, such as grammar and format problems, the proposed method not being verified by sea trials, and so on.The specific problems are as follows:
Question 1:In this article, for the first occurrence of these abbreviations “YOLO, UWB, VIO, NCC,…”,it needs to list the full name.In addition, there are many abbreviations in this paper, it is best to list them separately at the end of the article.
Question 2:In the “Abstract”, the authors present a long-term target anti-interference tracking (LTAT) method. Is the LATA method mentioned below the same as the LTAT method?It should be unified.
Question 3: The references in the current manuscript are relatively old and less. There is no current research on anti-interference control methods. The author is expected to refer to the following literature and add relevant papers in recent years.
Wang, WR, et al. Adaptive MPC trajectory tracking for AUV based on Laguerre function. Ocean Engineering, 2022, 261.
Wu, GX ; Luo, WJ ; Guo, JM ; Zhang, JW.A Sigmoid-plane adaptive control algorithm for unmanned surface vessel considering marine environment interference.Transactions of The Institute of Measurement and Control, 2022, 44(10):2076-2090.
Question 4:I think there was a grammatical error (“write”)on line 155.
Question 5:On line 158, the correlation coefficient matrix NCC(k) can be obtained by substituting Equation(5) into Equation(4)? I think it should be obtained by substituting Equation(2) into Equation(1).
Question 6:In this paper, why did the authors choosethe Cubature Kalman Filter (CKF) for optimization and prediction of the position? Have the authors clearly emphasized the strengths of the CKF?
Question 7:Here are some formatting issues.
(1) There is an extra period on line 311.
(2) In figure 9-15, the sequence number of the sub-graph should be unified with the full text into (a) or (b).
(3) Since there is only one picture in Figure 16, sothe letter “a” can be removed.
Question8:Does the author describe the severe sea conditions in detail during the simulation?
Question 9:Has the AUV been designed and manufactured, why there is no AUV-related parameter information, and why haven't the sea trialdata of the AUV been analyzed?If the algorithm proposed in the paper can be verifiedby sea trials, the engineering practicability of the algorithm can be proved.

Author Response

(The authors gave the same response as above.)

Reviewer 4 Report
The authors have developed a simulation for a Launch and Recovery system for AUVs using image processing. While this is an important research theme, this paper needs improvement.
The paper lacks a clear structure - a number of methods have been brought together without considering whether they are actually necessary. For example, the authors do not explanation the need for stereo cameras, and it seems that the system can work just fine with only YOLOv5 without the rest.
The physical setup of the system is not described. The errors described are absolute, but cannot be put in perspective without the physical dimensions of the AUV and LARS.
A large number of AUV recovery systems already in operation are not discussed by the authors.
Many of the authors claims - such as competitive, reliable, etc are not validated in the paper. Please carefully remove unnecessary parts.
Author Response
Dear Reviewer,
We feel great thanks for your professional review work on our article. As you are concerned, several problems need to be addressed. Responses to specific questions are uploaded in the documentation below.
We would like to take this opportunity to thank you for all your time involved and this great opportunity for us to improve the manuscript.
Sincerely,
The Authors

Round 2
Reviewer 1 Report
This paper has been effectively revised. The current version of the paper can be accepted.
Author Response
Dear Reviewer:
Thank you for your comments concerning our manuscript entitled"A Method for Long-term Target Anti-interference Tracking Combining Deep learning andCKF for LARS Tracking and Capturing." Those comments are precious and helpful for ourfuture research. We have studied the comments carefully and have made corrections whichwe hope will address the clarification needs. The following is our point-by-point reply to theeditorial comments.
We tried our best to improve the manuscript and made some changes marked in blue in therevised paper. Once again, thank you very much for your comments and suggestions
Sincerely.
Wenlin Yang
Guangdong Institute of Intelligent Unmannednstitution and address.System,
Guangzhou,511458. China
Telephone: +86-13909553296
Email: yangwenlin@gis.sia.cn
